# Energy Crops and Methane: Process Optimization of Ca(OH)_2_ Assisted Thermal Pretreatment and Modeling of Methane Production

**DOI:** 10.3390/molecules27206891

**Published:** 2022-10-14

**Authors:** Hasmet Emre Akman, Nuriye Altınay Perendeci, Can Ertekin, Osman Yaldiz

**Affiliations:** 1Department of Agricultural Machinery and Technology Engineering, Faculty of Agriculture, Akdeniz University, Antalya 07058, Turkey; 2Department of Environmental Engineering, Engineering Faculty, Akdeniz University, Antalya 07058, Turkey

**Keywords:** anaerobic digestion, methane, optimization, pretreatment, switchgrass

## Abstract

Switchgrass earned its place globally as a significant energy crop by possessing essential properties such as being able to control erosion, low cost of production, biomass richness, and appeal for biofuel production. In this study, the impact of a Ca(OH)_2_-assisted thermal pretreatment process on the switchgrass variety Shawnee for methane fuel production was investigated. The Ca(OH)_2_-assisted thermal pretreatment process was optimized to enhance the methane production potential of switchgrass. Solid loading (3–7%), Ca(OH)_2_ concentration (0–2%), reaction temperature (50–100 °C), and reaction time (6–16 h) were selected as independent variables for the optimization. Methane production was obtained as 248.7 mL CH_4_ gVS^−1^ under the optimized pretreatment conditions. Specifically, a reaction temperature of 100 °C, a reaction time of 6 h, 0% Ca(OH)_2_, and 3% solid loading. Compared to raw switchgrass, methane production was enhanced by 14.5%. Additionally, the changes in surface properties and bond structure, along with the kinetic parameters from first order, cone, reaction curve, and modified Gompertz modeling revealed the importance of optimization.

## 1. Introduction

Electrified transportation is expected to increase in the future; however, implementing its use will not happen everywhere, in every industry, and all at once [1]. Therefore, it is predicted that liquid biofuels will be extremely necessary for the ensuing decades. Sustainable and renewable fuel technologies have been focused on lignocellulosic energy crops as a promising substrate for producing biofuels. Due to enormously increased energy security demands coupled with climate change mitigation, the world’s biofuel production has also drastically expanded.

Energy crops, as a specific crop species, have been grown for advanced cellulosic biofuel production in recent decades. Switchgrass, one of these promising energy crops, belongs to the Poaceae family and is a native North American prairie grass species that can rapidly grow large amounts of carbohydrate-rich biomass. It is a species of C4-type plants (which utilize the Hatch–Slack cycle for carbon fixation), and as such, it can perform this process four times more productively than C3-type plants (which utilize the C3 pathway for carbon fixation). From an environmental point of view, switchgrass-based biofuels are carbon negative due to switchgrass’s high carbon sequestration ability owing to its deep and complex root system. Moreover, switchgrass has the ability to clean soils and waters by removing excess nutrients (N, P) originating from fertilization. This plant is also a great growth and living habitat for wildfowl and mammals, especially in ecotone zones between forest and agricultural fields [2,3]. From the agricultural perspective, switchgrass is very easy to establish and maintain due to its low nutrient and water requirements, high adaptability to many climates, and high resistance to naturally occurring pests and diseases [4]. Moreover, other advantages include expansion of the local market, new sources of income for the agricultural sector, diversification of feedstock sources, more effective resource utilization, innovation in industrial and agricultural technology, and economic growth [5]. The only drawback of switchgrass related to biofuel production bioprocessing is its lignocellulosic structure, which is highly resistant to hydrolysis [6].

Hydrolysis is the rate-limiting step for the anaerobic digestion (AD) process, even for less complex substrates than lignocellulosic biomass. Therefore, to utilize lignocellulosic biomass in AD efficiently, a pretreatment process needs to be applied. Pretreatment makes the crystalline, compact, and solid structures of lignocellulose suitable for chemical and biochemical conversion. A wide range of pretreatment processes, such as physical, chemical, thermal, and biological, and their combination has been investigated for lignocellulosic biomass in the literature for the last few years [7]. The objective of chemical pretreatment using chemicals is to break down the structure to open up cellulose and hemicellulose and remove lignin. Alkaline pretreatment efficiently removes the hemicellulose and lignin content by attacking the ether and ester bonds in lignocellulose [8]. The mechanism behind the alkaline pretreatment is explained to be the solvation and saponification of intermolecular ester bonds cross-linking xylan, lignin, and other hemicelluloses. The solvation and saponification eliminate these cross-links and enhance the porosity of the lignocellulosic materials [9]. The main advantage of alkaline pretreatment is the delignification [10]. Alkaline pretreatment depends on feedstock type, applied dose, and the pretreatment conditions [6]. CaO and Ca(OH)_2_ pretreatment is the most widely preferred among the various alkaline chemicals (sodium, potassium, calcium, and ammonium hydroxides) for enhancing the hydrolysis of various lignocellulosic biomasses [11,12,13,14] since it is less expensive and safer.

The AD of switchgrass, considering the methane yield for various harvesting seasons [15,16,17,18], co-digestion [19,20,21], AD kinetic modeling [22,23,24,25], and continuous AD with process performance [26,27,28,29], has been investigated in the literature. On the other hand, relatively limited research has been conducted on the pretreatment process to increase methane production from switchgrass [3,4,24,25,27,30,31,32]. Chemical-assisted thermal pretreatment application enhanced methane production in the range of 5–326% compared to raw switchgrass. Additionally, only three works in the literature [3,4,30] optimized the pretreatment of switchgrass and evaluated the process variables in detail to enhance methane production.

Since switchgrass is a complex substrate as lignocellulosic material, it needs pretreatment to break down the structure, open up cellulose and hemicellulose, and remove lignin to produce biogas. Therefore, this study aims to optimize the Ca(OH)_2_-assisted thermal pretreatment process to enhance methane production from the switchgrass variety, Shawnee. Central composite design (CCD) of response surface methodology (RSM) was selected for the optimization procedure. Solid loading as DM (3–7%), Ca(OH)_2_ concentration (0–2%), reaction temperature (50–100 °C), and reaction time (6–16 h) were chosen as independent variables. The effects of these independent variables on pH, sSugar, and methane production were investigated, and the Ca(OH)_2_-assisted thermal pretreatment process was modeled and optimized for maximum methane yield. The impact of the Ca(OH)_2_-assisted thermal pretreatment process on surface properties, molecular bond characterization of switchgrass, and anaerobic digestion kinetics were also evaluated.

## 2. Results and Discussion

### 2.1. Characterization Analysis Results

Characterization of biomass for biofuel production is the first step for evaluation. A field-dried, milled, and homogenized variety of Shawnee as an energy crop was used to characterize biomass. The general composition of switchgrass Shawnee is presented in Table 1, and detailed analysis results can be found in Başar et al. [30]. The variety of Shawnee contained 93.8% dry matter, indicating a moisture content of approximately 6.2%. Volatile solids (VS) were measured as 82.4%, mostly formed of lignocellulose. The lignocellulose composition of switchgrass is quantified as 34.8% hemicellulose, 33.1% cellulose, 26.5% neutral detergent solubles, and 5.6% lignin. Lemusa et al. [33] found 32.6% hemicellulose, 36.8% cellulose, and 6.30% lignin in the switchgrass sample. The results from this study are very close to Lemusa et al.’s [33] findings. Niu et al. [34] and Imam and Capareda [35] have also reported similar results. The elemental composition of the field-dried variety of Shawnee was quantified as 40.13% C, 5.75% H, and 0.87% N, representing switchgrass as a convenient energy crop for biofuel production. Elemental analysis results are in line with the values measured for the switchgrass variety of Shawnee in literature [35,36,37,38]. Furthermore, structural carbohydrate (polymeric carbohydrates, namely cellulose and hemicellulose. Glucose, mannose, galactose, xylose, and arabinose are the monosaccharide component of the cellulose and hemicellulose) analysis resulted that switchgrass variety of Shawnee has 29.41% glucose and 17.36% xylose. The results of the structural carbohydrate analysis were also consistent with the literature [39,40].

### 2.2. Effect of Ca(OH)_2_ Assisted Thermal Pretreatment Process on Switchgrass

Thomas et al. [41] indicated that alkaline pretreatments have a variety of effects, such as (i) delignification, which results in the unmasking of cellulose, (ii) the breakdown of hemicellulose and cellulose, and (iii) a reduction in crystalline cellulose. pH, sSugar, and BMP analyses were carried out on raw and pretreated Shawnee switchgrass to observe the pretreatment effect. pH, sSugar, and BMP results of Ca(OH)_2_-assisted thermal pretreatment experiments are shown in Figure 1a–c, respectively.

Since the Ca(OH)_2_ is applied in the range of 0–2% in the pretreatment experiments, it is expected that changes in the pretreatment inlet and outlet pHs depend on the applied amount of Ca(OH)_2_. As can be seen in Figure 1a, the pH values of the variety of Shawnee switchgrass that were not pretreated with Ca(OH)_2_ at the reaction temperatures of 50, 75, and 100 °C were measured close to each other before and after the pretreatment experiments. While initial pH values were measured as 6.5–6.7, pH values varied between 5.2 and 6.7 depending on solid loading and reaction time after the pretreatment. In the pretreatment experiments assisted with 2% Ca(OH)_2_, the initial pH values ranged from 10.3 to 10.9. At the end of the reactions, at low temperatures (50 °C), pH values changed in the range of 7.7–8.2 depending on the pretreatment reaction time, while at high temperatures (100 °C), it changed between 6.5 and 6.9 regardless of the reaction time. When 1% Ca(OH)_2_ was applied in the pretreatments, the initial pH values were 9.5–9.8, while pH values were measured as 6.1–6.8 after the pretreatment. The pH values were observed to be lower at high temperatures.

sSugar analysis found that raw switchgrass has 50 mg sSugar VS^−1^. The amounts of sSugar were measured at 7% solid loading (SL), as DM is higher than 3% SL when 50 °C reaction temperature, 6 h and 16 h reaction times, and without Ca(OH)_2_ conditions are applied in the pretreatments. It is normal that sugar’s solubilization from high solid loading was higher than from low solid loading. On the other hand, under the pretreatment conditions of 50 °C reaction temperature, 6 h and 16 h reaction times, and 2% Ca(OH)_2_, the amount of sSugar at 7% solid loading was lower than 3% solid loading. Higher sSugar values were measured from the raw switchgrass samples at all conditions of 75 °C reaction temperature applied. Furthermore, at 100 °C reaction temperature conditions, the sSugar values measured at 7% SL were higher than those measured at 3% solid loading.

Raw Shawnee switchgrass produced 217.1 mL CH_4_ gVS^−1^. In the literature, the methane production potential of switchgrass was reported as 169–252 mL CH_4_ gVS^−1^ by Masse et al. [18] and 104.0–153.5 mL gVS^−1^ by Frigon et al. [16] depending on different harvesting seasons. Furthermore, the methane potential of raw Kanlow was reported as 197.39 mL CH_4_ gVS^−1^ by El-Mashad et al. [24], the raw Alamo variety was presented as 246 mL CH_4_ gVS^−1^ by Grigatti et al. [42], raw Kanlow was found 215.5 mL CH_4_ gVS^−1^ by Başar et al. [30], and raw Shawnee was quantified as 208.4 mL CH_4_ gVS^−1^ by Başar and Perendeci [3]. Considering the results in previous studies, obtained results from this study are in the range of reported methane yields.

The maximum methane generation was achieved as 231.4 mL CH_4_ gVS^−1^, specifically under the pretreatment conditions of 100 °C reaction temperature, 16 h of reaction time, 0% Ca(OH)_2_, and 3% solid loading. Methane generation was enhanced by 6.6% compared to raw switchgrass. In contrast, the lowest methane production was observed as 62.4 mL CH_4_ gVS^−1^ when the 75 °C reaction temperature, 11 h reaction time, 2% Ca(OH)_2_, and 5% solid loading pretreatment conditions.

As is seen in Figure 1c, an increase in solid loading from 3% to 7% led to a severe fall in BMP results under all conditions. Zheng et al. [21] observed a similar outcome that mono-digestion of a high dose (>4% TS) of switchgrass led to volatile fatty acid accumulation and process failure, suggesting that a high solid loading might introduce excessive quantities of easily digestible material into digesters.

When 50 °C reaction temperature, 6 h and 16 h reaction times, and 3% solid loading conditions are applied in the pretreatments, an increase in Ca(OH)_2_ from 0% to 2% resulted in high BMP values. A similar trend was observed at 100 °C reaction temperature, 6 h reaction time, and 3% solid loading conditions, but low BMP production was obtained when raising the reaction time from 6 h to 16 h. Furthermore, BMP values slightly rose or remained stable while the reaction temperature increased from 50 °C to 100 °C. When a reduction in methane production is observed, it should be kept in mind that process optimization is required for the amount of chemicals used in the pretreatment process, depending on the composition of lignocellulose. Gunerhan et al. [43] indicated that the ability to optimize the process within a significant design boundary and assess the positive and negative effects of independent variables and their ranges are compelling qualities in selecting a wide range of independent variables.

### 2.3. Modeling and Optimization of Ca(OH)_2_ Assisted Thermal Pretreatment Process

BMP results from the experiments of 51 Ca(OH)_2_-assisted thermal pretreatment, containing duplicates of 25 varied pretreatment conditions and triplicate of the center point, were loaded into Design Expert software. The model for BMP was developed by RSM. The accuracy of the BMP model was evaluated by the highest determination of coefficient (*R*^2^), adjusted-*R*^2^, and predicted-*R*^2^. In addition, modification (*Backward*) was applied to the quadratic BMP model to obtain high *R*^2^. Backward selection seeks to remove terms from a model that are detrimental to the criterion. *p*-value criterion was used in the backward selection. The BMP model equation was used to optimize the Ca(OH)_2_-assisted thermal pretreatment process for maximum methane production with an acceptable process cost. BMP Model equation, statistics, and information are presented in Table 2.

A BMP model with *R*^2^ of 0.8572, adjusted–*R*^2^ of 0.8121, and predicted–*R*^2^ of 0.7646 was developed. The coefficient of determination, expressed as *R*^2^, shows the model’s predictive power in percent. The developed model could only explain 85.72% of the relation between BMP and independent variables. The quadratic modified BMP model presented a low probability result (*p*-value < 0.0001), indicating the model is in the confidence range of 99%. *p*-values less than 0.05 indicate that model terms are significant. According to the BMP model statistics, solid loading (*B*) and Ca(OH)_2_ concentration (*D*) were found to be significant terms from the first-degree effect terms, whereas reaction time (*C*) and reaction temperature (*A*) were determined insignificant terms in the BMP model of Ca(OH)_2_-assisted thermal pretreatment process. Furthermore, solid loading * Ca(OH)_2_ concentration (*BD*), square terms of solid loading (*B*^2^), and reaction time (*C*^2^) are effective independent variables in the BMP model. Since the first-degree effect of the reaction temperature (*A*) does not change BMP directly and to obtain a hierarchical model, insignificant terms from the quadratic BMP model were excluded by the application of backward modification.

The individual effects of independent variables of Ca(OH)_2_ assisted thermal pretreatment process on BMP and desirability (0.977) are presented in Figure 2. The individual effect of each independent variable on the BMP model presents that 179–236 mL CH_4_ gVS^−1^ should be produced by the application of 3% solid loading, 6 h reaction time, and 0% Ca(OH)_2_ concentration conditions in the pretreatment process. As is seen in Figure 2, the BMP value is observed to be stable between the 50 and 100 °C reaction temperature. On the other hand, BMP values decrease from 231.4 to 81.6 mL CH_4_ gVS^−1^ by increasing solid loading from 3% to 7%. Furthermore, increasing solid loading, reaction time, and Ca(OH)_2_ concentration resulted in low desirability and BMP values.

3D response surface plots for the effects of independent variables on the BMP model are given in Figure 3a–d. In the interaction effect of solid loading and reaction temperature presented in Figure 3a, the maximum BMP value was reached only at the lowest solid loading of 3%. The graph also indicates that temperature change did not affect the BMP value where solid loading is kept constant, but the drops in solid loading value remarkably caused a rise in BMP values.

In Figure 3b, the maximum BMP value is also reached only at the lowest solid loading value of 3%. Furthermore, keeping the solid loading at a constant 7% and increasing the Ca(OH)_2_ concentration positively impacted BMP values. Where solid loading is kept at a constant minimum of 3%, increasing the Ca(OH)_2_ concentration did not show any significant impact on the methane yield. On the other hand, where Ca(OH)_2_ concentration was kept at its constant, increasing the solid loading adversely affected the BMP values; specifically, at the maximum solid loading and 0% Ca(OH)_2_ concentration, BMP resulted in the lowest value. Figure 3c revealed that, within this experimental design, the optimum reaction time is around 6 and/or 16 h to reach the highest methane yield. Between 6 and 16 h reaction time, methane production yield is observed to have a decreasing trend. Figure 3d shows that a change in reaction temperature and Ca(OH)_2_ concentration in any direction did not affect the BMP values significantly. Conclusively, solid loading * Ca(OH)_2_ concentration was the most effective interaction for the maximum methane yield.

Optimization of a Ca(OH)_2_-assisted thermal pretreatment process was implemented to generate the maximum specific methane yield using the BMP model. In the optimization solid loading (%) (+++++), reaction time (h) (+++++), Ca(OH)_2_ concentration (%) (+++++) were minimized, whereas reaction temperature (°C) (+++++) and BMP yield (mL CH_4_ g VS^−1^) (+++++) was maximized. Under these conditions, optimum Ca(OH)_2_-assisted thermal pretreatment process conditions were found as 3% solid loading, 100 °C reaction temperature, 0% Ca(OH)_2_ concentration, and 6 h reaction time with the 0.977 value of highest desirability. The BMP model predicted methane yield as 213.5 mL CH_4_ gVS^−1^ under optimized conditions. The Ca(OH)_2_-assisted thermal pretreatment trial was done under optimized conditions for the validation. Pretreated switchgrass was subjected to the BMP test, and methane yield was acquired as 248.7 mL CH_4_ gVS^−1^. Enhancement of methane yield was computed as 14.5%, compared to raw switchgrass under optimized validation conditions.

Most of the work in the literature focused on the production of methane from switchgrass without pretreatment. Pretreatment conditions for switchgrass and results available in the literature are presented in Table 3.

It can be concluded from Table 3 that there is limited work focused on the pretreatment of switchgrass for methane production in the literature. Pretreatment work can be classified into two groups. Chemical pretreatment works are supported by the application of heat. Basic (Ca(OH)_2_, NaOH, KOH), acidic (HCl, H_2_SO_4_), and oxidizing (H_2_O_2_) chemical agents were preferred in pretreatment. As seen in Table 3, with the pretreatment application, the methane production was 12.2–325.9% enhanced compared to untreated switchgrass [3,4,24,27,31]. Thermal pretreatment as microwave and steam explosion also enriched methane production by 5.2–9.1% [25,26,27,28,29,30,31,32]. Results show that combined chemical and thermal pretreatment application seems to be more effective. The enhancement result obtained in this study is in line with the literature. On the other hand, this work is the first to focus on optimizing Ca(OH)_2_-assisted thermal pretreatment, considering all the process variables for methane enhancement. However, this result revealed that Ca(OH)_2_-assisted thermal pretreatment for switchgrass does not significantly affect methane enhancement.

### 2.4. Effects of Ca(OH)_2_-Assisted Thermal Pretreatment Process on Switchgrass Surface Modification and Molecular Bond Changes

SEM images and the FTIR analysis of the raw and pretreated switchgrass samples were performed to appraise the alterations in surface properties and the structural-chemical modifications that emerged during the Ca(OH)_2_-assisted thermal pretreatment process. SEM images of raw switchgrass sample (a), pretreated switchgrass samples at 3% SL, 50 °C, 0% Ca(OH)_2_, 6 h, (b), 3% SL, 100 °C, 2% Ca(OH)_2_, 16 h (c), and 3% SL 100 °C, 0% Ca(OH)_2_, 6 h (d) are presented in Figure 4.

In the SEM image of the raw switchgrass, the sample surface is smooth, compact, and rigid (Figure 4a). Therefore, there is no deformation noticed naturally. On the other hand, the SEM image in Figure 4b reveals that the pretreatment achieves peeling off the outer crust as a function of reaction temperature and time. Additionally, sub-surface canals are observed to begin deterioration. The SEM image of the pretreated switchgrass sample at 3% SL, 2% Ca(OH)_2_, 16 h, and 100 °C conditions (Figure 4c) indicates how application caused the entire breakdown of the outer crust and sub-surface canals. As a result, the porosity of the material structure has increased, thus enhancing further hydrolysis. Notably, in these conditions, the fibril structure of the switchgrass was entirely altered, and structural unity was distorted. The SEM image of the switchgrass sample at the optimized pretreatment conditions (3% SL, 0% Ca(OH)_2_, 6 h, and 100 °C) (Figure 4d) shows that the outer crust broke down totally, and slight deterioration is observed on the sub-surface canals.

FTIR spectra of raw switchgrass sample (a), pretreated switchgrass samples at 3% SL, 50 °C, 0% Ca(OH)_2_, 6 h, (b), 3% SL, 100 °C, 2% Ca(OH)_2_, 16 h (c), and 3% SL 100 °C, 0% Ca(OH)_2_, 6 h (d) are presented in Figure 5. Bands in FTIR spectra available in the literature are also given in Table 4 for comparison.

Spectroscopic results obtained by FTIR analysis indicate that raw and pretreated switchgrass samples show similar binding properties at the wavelength between 400 and 1800 cm^−1^. It is observed that the switchgrass sample pretreated at 3% SL, 100 °C, 2% Ca(OH)_2_, and 16 h conditions show sectional similarities with the switchgrass sample pretreated at 3% SL, 100 °C, 0% Ca(OH)_2_, and 6 h conditions. Nevertheless, upon careful examination, differences in chemical bonds were observed in both samples. Between the 400–1800 cm^−1^ wavelength, 470 cm^−1^, 560 cm^−1^, 620 cm^−1^, 775 cm^−1^, 895 cm^−1^, 1050 cm^−1^, 1180 cm^−1^, 1245 cm^−1^, 1440 cm^−1^, 1465 cm^−1^, 1580 cm^−1^ and 1735 cm^−1^ wavelengths in particular show remarkable differences. FTIR spectrum of raw switchgrass sample shows differences in Si-O-Si and (PO_4_)_3_^−^ bonds at 470 cm^−1^ wavelengths compared to pretreated switchgrass samples. Additionally, a notable resonance in Si-O-Si and (PO_4_)_3_^−^ bonds is observed for the switchgrass samples pretreated at 3% SL, 100 °C, 2% Ca(OH)_2_, and 16 h conditions compared to raw switchgrass, indicating a concentration increase of fragments containing these groups. However, the resonance in C-O and C=O bonds at 620 cm^−1^ wavelengths remained identical when high temperatures were applied to the switchgrass samples. Thus, this shows how temperature application makes a difference in C-O and C=O bonds. 775 cm^−1^ wavelength corresponds to resonance in NH_2_ bonds. Switchgrass samples pretreated at 3% SL, 100 °C, 2% Ca(OH)_2_, and 16 h conditions have minimum absorbance value compared to raw switchgrass samples. This observation indicates how much deterioration occurred under the effects of Ca(OH)_2_-assisted temperature pretreatment. Moreover, 895 cm^−1^ wavelength states the resonance in β-glucosidic bonds directly related to cellulose. Switchgrass sample either pretreated at 3% SL, 50 °C, 0% Ca(OH)_2_, 6 h, or at 3% SL, 100 °C, 0% Ca(OH)_2_, 6 h conditions showed similar properties. However, the sample pretreated at 3% SL, 100 °C, 2% Ca(OH)_2_, and 16 h conditions had a low absorbance value. This is an indication of the deterioration of the β-glycosidic bonds. The 1050 cm^−1^ wavelength indicates C-O, C=C, C-OH, and C-O-C stretching vibrations in cellulose and hemicellulose compounds. FTIR spectrums of raw and pretreated switchgrass samples at 3% SL, 50 °C, 0% Ca(OH)_2_, 6 h, and 3% SL, 100 °C, 0% Ca(OH)_2_, 6 h conditions indicated similar absorbance values. However, a minimum absorbance value was obtained from the switchgrass sample pretreated at 3% SL, 100 °C, 2% Ca(OH)_2_, and 16 h conditions. This shows the tension in the C-O, C=C, C-OH, and C-O-C bonds of cellulose and hemicellulose. Alteration in the band position and intensity may represent a reduction in the content of the structural ingredients by the application of pretreatment [44]. The peaks at 1180 cm^−1^ and 1245 cm^−1^ wavelengths are related to the C-O-C asymmetrical bending vibration in cellulose and hemicellulose compounds and the C-O bonds of acetyl groups of hemicellulose, respectively. Switchgrass samples pretreated at 3% SL, 100 °C, 2% Ca(OH)_2_, and 16 h conditions resulted in lower absorbance values than others at these wavelengths. This is evidence of the disintegration of the acetyl groups of hemicellulose. In the fingerprint zone of the spectra, modifications in spectra, especially at 1440, 1465, and 1580 cm^−1^ wavelengths, can be observed from the pretreated switchgrass samples compared to raw switchgrass. Chemical bonds at 1140, 1465, and 1580 are related to lignin decomposition. Out of the fingerprint region, at 2900 and 3400 cm^−1^ wavelengths, modifications were monitored that indicate an alteration in C-H and O-H bonds.

**Table 4 molecules-27-06891-t004:** Bands in FTIR spectra available in the literature.

Wavelength (cm^−1^)	Affected Chemical Bond and Impact	Reference
470	Resonance in bonds of Si-O-Si and (PO_4_)^3−^	[45,46]
620	Resonance in C-O, C=O bonds	[47]
775	Resonance in NH_2_ bonds	[48]
895	Resonance in β-glucosidic bonds, Indicator for crystalline and amorphous cellulose rate	[49,50,51]
1050	C-O, C=C, C-OH, and C-O-C tensions in cellulose and hemicellulose	[52,53]
1180	Asymmetrical C-O-C tension in cellulose and hemicellulose	[54]
1245	C-O adsorption of acetyl groups in hemicellulose	[52]
1280	C-H warping in crystallized cellulose	[55]
1440	O-H bond in linear warping of hemicellulose and lignin	[54]
1465	C-H deformation in lignin	[54]
1580	Resonance of aromatic rings in lignin	[51]
1735	Resonance of bonds in ketone and ester carbonyl groups	[55]
2900–3400	Resonance and tension of C-H and O-H bonds in cellulose	[51,53,56]

### 2.5. Kinetic Modeling

The BMP test was finalized at 56 days. Since the biogas production reached a plateau after the first 28 days, this period was considered in the kinetic modeling. Predicted methane production and kinetic parameters from the first order, cone, modified Gompertz, and reaction curve modeling, along with the statistical indicators as *R*^2^ and Adjusted-*R*^2^, are presented in Table 5. Simulated methane potential values were plotted against experimental data to visually compare, shown in Figure 6.

High compatibility (*R*^2^) was achieved for all models. All four models were successfully applied to determine the biogas potential of the switchgrass. The hydrolysis rate constant (*k*) is an important parameter showing the methane production efficiency in the anaerobic degradation process. The *k* values estimated with the first order kinetic model for the different pretreatments applied to the switchgrass varied in the range of 0.115–0.159 d^−1^. The *k* value varies between 0.136 and 0.210 d^−1^ from cone modeling. Parallel with the first order modeling, the minimum *k* value was estimated from the pretreated switchgrass at 3% SL, 50 °C, 0% Ca(OH)_2_, 6 h and was followed by the pretreated switchgrass at 3% SL, 100 °C, 0% Ca(OH)_2_, 6 h conditions and the raw sample. The estimated and experimental results show a variance between 2.55% and 6.41%. The high *R*^2^ values and minor differences between model results and experimental results for methane production indicate that the cone model can be used successfully.

Modified Gompertz modeling results indicate the maximum *R_m_* value would be 17.44 mL CH_4_ gVS^−1^ d^−1^ for pretreated switchgrass at 3% SL, 100 °C, 2% Ca(OH)_2_, 16 h conditions, 15.97 mL CH_4_ gVS^−1^ d^−1^ for raw sample, 14.37 mL CH_4_ gVS^−1^ d^−1^ from pretreated switchgrass at 3% SL, 100 °C, 0% Ca(OH)_2_, 6 h, and 14.25 mL CH_4_ gVS^−1^ d^−1^ from pretreated switchgrass at 3% SL, 50 °C, 0% Ca(OH)_2_, 6 h. Low alpha values are consistent with experimental BMP results. BMP production started quickly in all four samples.

When examining the results of the reaction curve model, the *Rm* value was calculated as 26.33 mL CH_4_ gVS^−1^ d^−1^ for pretreated switchgrass at 3% SL, 100 °C, 2% Ca(OH)_2_, 16 h conditions, as in the modified Gompertz model, followed by 22.70 mL CH_4_ gVS^−1^ d^−1^ for raw, 20.33 mL CH_4_ gVS^−1^ d^−1^ from pretreated switchgrass at 3% SL, 100 °C, 0% Ca(OH)_2_, 6 h, and 20.19 mL CH_4_ gVS^−1^ d^−1^ from pretreated switchgrass at 3% SL, 50 °C, 0% Ca(OH)_2_, 6 h.

In conclusion, in evaluating the success of each four models in this study, every model achieves applicable results, while first order, cone, and reaction curve methods proved to have a better prediction capacity than modified Gompertz. For the comparison, the effects of pretreatments on methane production kinetic parameters available in the literature are presented in Table 6.

Elalami et al. [7] concluded that the olive pomace maximum methane production rate (*Rm*) and ultimate specific methane production were improved. At the same time, lag phase time (*λ*) decreased due to the application of NaOH pretreatment (%2–4–8, 25–50 °C, and 1–2 days), except the lag phase time (*λ*) for the condition of 8% NaOH, 50 °C and 1 day. Shen et al. [31] applied chemical pretreatments with KOH, NaOH, and Ca(OH)_2_ to the switchgrass. The *λ* and maximum methane production rate (*μm*) values were calculated as 0.8 d and 6.3 mL CH_4_ gVS^−1^ d^−1^, respectively, for 5% Ca(OH)_2_ pretreatment conditions.

Compared to the untreated raw switchgrass (*λ: 3.1 d* and *μm: 1.9* mLSTP gVS^−1^ d^−1^), while *λ* decreased, the *μm* value increased by the pretreatment. In this study, parallel with the literature, the *Rm* value increased, and the λ value decreased obtained from the reaction curve and modified Gompertz models compared to raw by applying 3% SL, 100 °C, 2% Ca(OH)_2_, 16 h pretreatment condition.

Jackowiak et al. [25] calculated the *k* value as 0.095–0.134 d^−1^ for microwave-pretreated switchgrass at different temperatures (90–180 °C). Wu et al. [32] also found *k* values between 0.021–0.075 d^−1^ for the leaf and 0.021–0.070 d^−1^ for the stem part of the switchgrass. Obtained *k* values from this study are slightly higher than the literature values. However, it should be noted that *k* values were calculated for the first time for Ca(OH)_2_-assisted thermal pretreatment for switchgrass.

The cone model uses the kinetic parameters of the hydrolysis rate constant *k* and the shape factor *n* to predict methane production Unyay et al. [26] calculated the k and n values as 0.067–0.080 and 1.7–1.9, respectively, for the anaerobic digestion of raw switchgrass at different substrate–inoculum ratios. As a result of the Ca(OH)_2_-assisted thermal pretreatment in this study, *k* constants increased, and the n values decreased compared to the work of Unyay et al. [26].

## 3. Materials and Methods

### 3.1. Switchgrass

In this study, the switchgrass variety of Shawnee is utilized as biomass. Switchgrass was planted and dried on the field in Karapınar (Konya, Turkey). The size of the samples was reduced to 3 mm to achieve textural homogeneity and consistency [57,58,59,60]. Samples of Shawnee were kept in labware plastic bags at 20 °C temperature for further experimentation for one year.

### 3.2. Switchgrass Characterization

Total solids (TS), volatile solids (VS) [61], Van Soest detergent fiber [62], structural carbohydrate [63], and elemental analysis were performed to characterize the composition of switchgrass. Measurement of C, H, N, and S content of switchgrass was performed by LECO CHNS-932 Elemental Analyzer (LECO Corporation, St. Joseph, MI, USA). Structural carbohydrate analysis was performed with Thermo Scientific UltiMate^®^ 3000 HPLC (Thermo Fisher Scientific, Waltham, MA, USA) with an Aminex HPX-87P column (Bio-Rad Laboratories Inc., Hercules, CA, USA) for characterization of switchgrass.

### 3.3. Experimental Design and Analysis

The experimental design of Ca(OH)_2_-assisted thermal pretreatment on switchgrass was applied by Central Composite Design (CCD) and Response Surface Method (RSM). CCD with 3 levels of 4 independent variables as switchgrass solids loading (3–7%) as DM, reaction temperature (50–100 °C), Ca(OH)_2_ concentration (0–2%), and reaction time (6–16 h) were designated for the process optimization. Biochemical methane potential (BMP) and sSugar results were selected as dependent variables. Design Expert 11 software (Stat-ease Inc., Minneapolis, MN, USA) was used for experimental design and statistical analysis.

Experimental results were modeled, and the obtained regression model was evaluated by the analysis of variance (ANOVA), regression coefficients, and the *p*- and F-Values. Model quality was determined by the coefficient of determination (*R*^2^) and an adjusted determination coefficient (*adj-R*^2^). Optimization of Ca(OH)_2_-assisted thermal pretreatment was done by using maximization and minimization criteria for independent and dependent variables [64,65] by using obtained model equation. The optimization module investigated the merging of independent variable values that meet the demands placed on each of the dependent variables. The optimizations were performed for maximum methane production potential.

### 3.4. Ca(OH)_2_-Assisted Thermal Pretreatment

Ca(OH)_2_-assisted thermal pretreatment experiments were conducted in 1 L laboratory-scale glass reactors equipped with a reflux condenser immersed in a 2 L oil heating bath. To provide the desired temperature and mixing conditions, magnetic stirrers with heaters (IKA C-Mag HS-7, IKAWerke GmbH, Staufen, Germany) and oil baths were used. Glass reactors were closed with condensers, which were kept at constant temperature by a circulating water bath (Lauda Alpha RA-8, Lauda Dr. R. Wobser GmbH, Lauda-K€onigshofen, Germany) to avoid evaporation.

In the Ca(OH)_2_-assisted thermal pretreatment experiments, reactors were heated to predetermined temperatures after the addition of a calculated amount of switchgrass, Ca(OH)_2_, and deionized water. The reaction time was started, and the temperature was kept at the determined temperature during the pretreatment experiments. The reactor contents were continuously mixed with magnetic stirrers at 150 rpm. All pretreatment experiments were performed in duplicates. At the end of reaction time, the temperature of the reactor content was rapidly cooled down to room temperature. pH values of reactor content were measured before and after the pretreatment experiments. Reactor contents were centrifuged two times at 4400 rpm for 5 min and 14,500 rpm for 15 min to obtain the liquid phase for soluble sugar (sSugar) analysis. Soluble sugar (sSugar) analysis was performed by the Dinitro Salicylic Acid (DNS) method [66] to observe the pretreatment efficiency. Mixed solid and liquid samples for BMP tests were kept at −20 °C.

### 3.5. Biochemical Methane Potential Test (BMP)

The standard BMP test was applied to define the biochemical methane potentials of raw and pretreated switchgrass by Ca(OH)_2_-assisted thermal. The BMP experiment works on the basis of incubating a specific amount of switchgrass samples mixed with anaerobic sludge at 35 °C and detecting the gas composition and volume produced. The BMP test was conducted according to Perendeci et al. [67]. BMP experiments were performed in 500 mL glass reactors with a 400 mL working volume. pH was set to neutral, and reactors were flushed with N_2_/CO_2_ (70%/30%) gas blend for the anaerobic condition after the addition of the switchgrass sample, seed sludge, required nutrients, buffer solution, and deionized water. The substrate-to-seed ratio was selected as 0.5 (gVS gVS^−1^ for solid samples). BMP reactors were incubated at 35 °C, and experiments were finalized at 56 days. The gas composition of N_2_, CH_4_, and CO_2_ in headspaces of the BMP reactors was quantified by Varian CP-4900 Micro-Gas Chromatography (Varian Inc., Palo Alto, CA, USA). Details of the measurements of biogas volume and the analysis of gas composition can be found in Basar et al. [30].

### 3.6. Scanning Electron Microscopy (SEM) and Fourier Transform Infrared Spectroscopy (FTIR)

The surface structure of raw and pretreated switchgrass samples was evaluated by SEM to determine the changes caused by Ca(OH)_2_-assisted thermal pretreatment. Samples firstly were lyophilized and coated with gold-palladium for 120 s under 18 mA current (Polaron SC7620 Sputter Coater, Quorum Technologies, Lewes, UK). After coating, the samples were monitored at different magnifications, and images were kept with SEM (Zeiss Leo 1430, Zeiss Group, Oberkochen, Germany) at 15 kV voltage.

To determine the effects of Ca(OH)_2_-assisted thermal pretreatment on the molecular bond characteristics of the lignocellulosic structure, Fourier transforms infrared spectroscopy (FTIR) analyzes were performed for both raw and pretreated samples. Samples were lyophilized, and measurements were performed by a Perkin Elmer, Spectrum Two in 400 cm^−1^ and 4000 cm^−1^ wavelength ranges.

### 3.7. Kinetic Modeling

To evaluate the effect of Ca(OH)_2_-assisted thermal pretreatment on the anaerobic digestion reaction kinetics, specific cumulative methane production was modeled by using modified Gompertz and reaction curve equations. Furthermore, first order and cone models were implemented for the determination of the hydrolysis rate. Detailed descriptions of the model equations can be found in the work of Yılmaz et al. [68]. All model simulations were done by AQUASIM 2.1 Trial version.

## 4. Conclusions

Energy crops have garnered considerable attention due to their significant potential as biofuel feedstock and agricultural benefits. Most switchgrass varieties are a powerful alternative to produce energy-efficient biofuel. In this study, Ca(OH)_2_-assisted thermal pretreatment was applied to energy crops to enhance biodegradability and biofuel yield and remove a significant handicap of lignocellulose by unmasking cellulose. Neat optimization revealed that solid loading as DM (3–7%) and Ca(OH)_2_ concentration (0–2%) were found to be significant independent variables compared to reaction temperature (50–100 °C) and time (6–16 h) within range desing boundary. Essentially, increasing solid loading, reaction time, and Ca(OH)_2_ concentration resulted in low desirability and methane production yield. A biochemical methane production model was built and validated specific methane yield with 14.5% increases compared to untreated switchgrass. Nevertheless, it is concluded that Ca(OH)_2_-assisted thermal pretreatment did not significantly affect methane enhancement compared to other pretreatment methods.

## Figures and Tables

**Figure 1 molecules-27-06891-f001:**
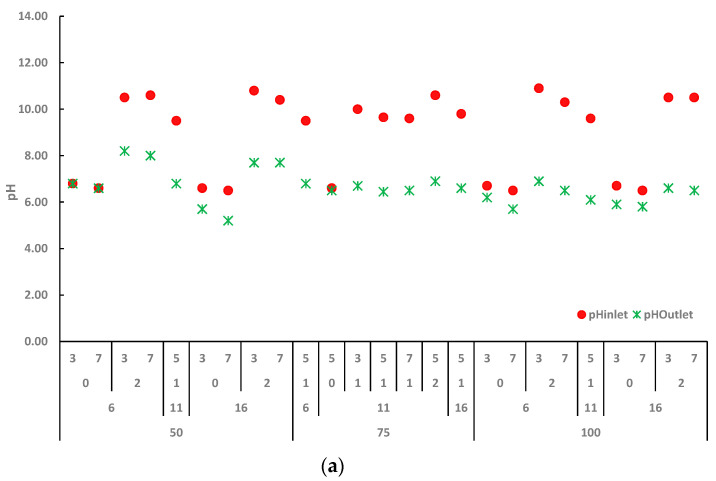
Ca(OH)_2_-assisted thermal pretreatment effects on pH (**a**), sSugar (**b**) and BMP (**c**).

**Figure 2 molecules-27-06891-f002:**
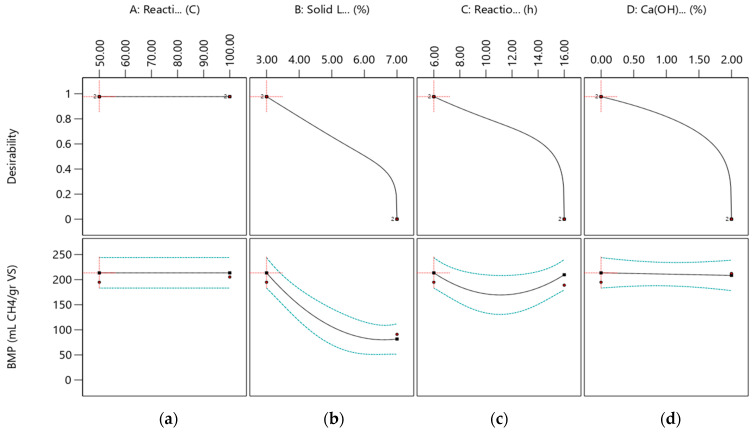
Desirability and BMP prediction values related to independent process variables as reaction temperature (**a**), solid loading (**b**), reaction time (**c**), and Ca(OH)_2_ concentration (**d**).

**Figure 3 molecules-27-06891-f003:**
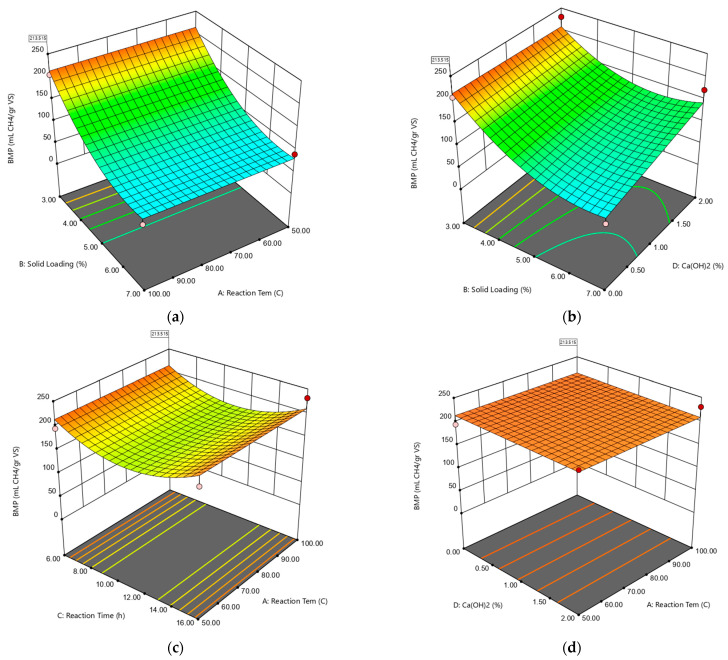
3D response surface plots for the effects of independent variables on BMP. The response surface diagrams: (**a**) Solid loading-Reaction time, (**b**) Solid loading-Ca(OH)_2_ concentration, (**c**) Reaction time-Reaction temperature, (**d**) Ca(OH)_2_ concentration-Reaction Temperature.

**Figure 4 molecules-27-06891-f004:**
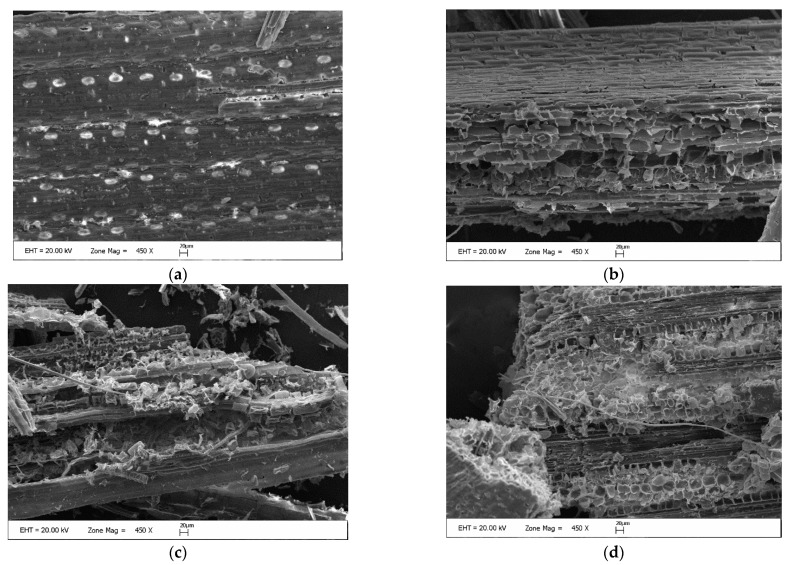
SEM images of raw switchgrass sample (**a**), Pretreated switchgrass sample 3% SL, 50 °C, 0% Ca(OH)_2_, 6 h, (**b**) Pretreated switchgrass sample 3% SL, 100 °C, 2% Ca(OH)_2_, 16 h (**c**), and Pretreated switchgrass sample 3% SL, 100 °C, 0% Ca(OH)_2_, 6 h (**d**).

**Figure 5 molecules-27-06891-f005:**
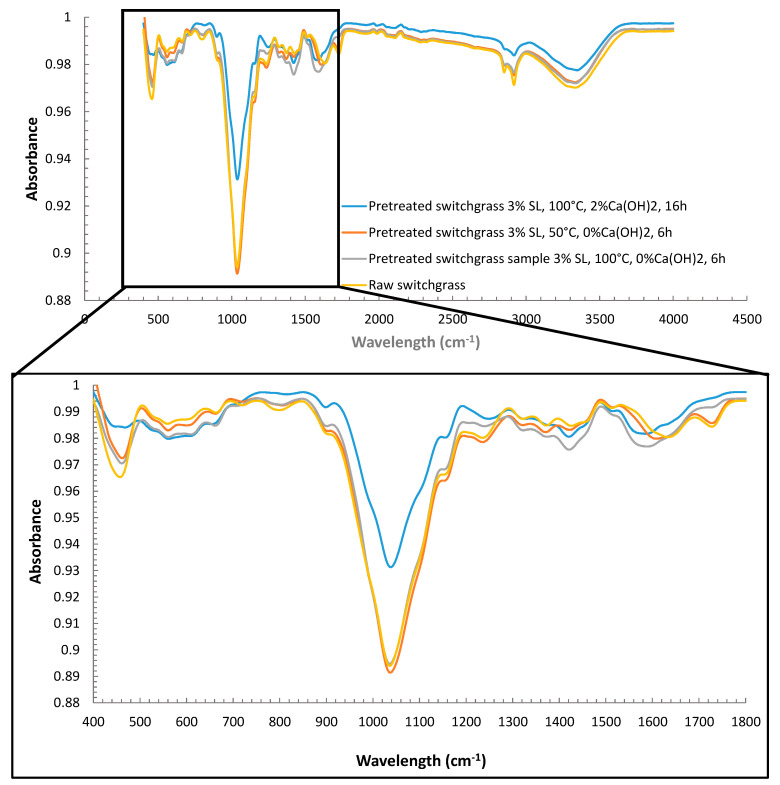
FTIR spectra of raw switchgrass sample, Pretreated switchgrass sample 3% SL, 50 °C, 0% Ca(OH)_2_, 6 h, Pretreated switchgrass sample 3% SL, 100 °C, 2% Ca(OH)_2_, 16 h, and Pretreated switchgrass sample 3% SL, 100 °C, 0% Ca(OH)_2_, 6 h.

**Figure 6 molecules-27-06891-f006:**
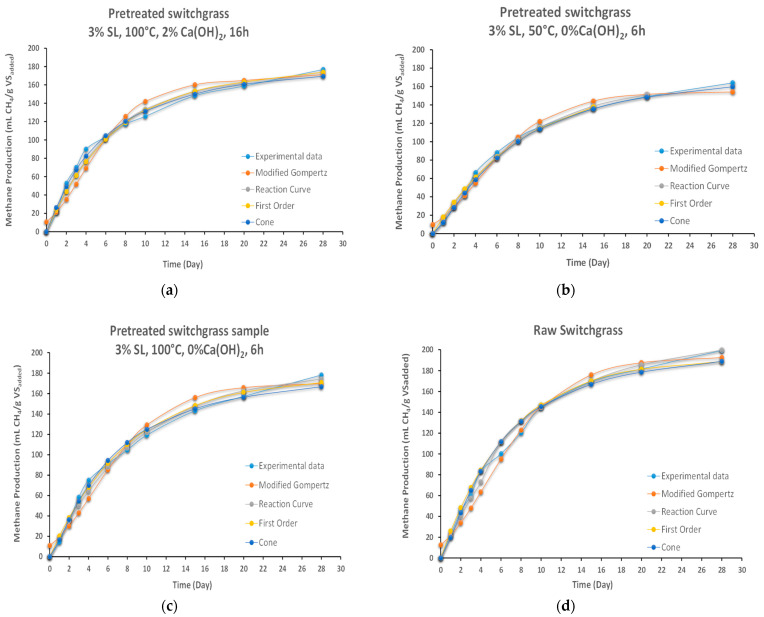
Model predictions and experimental methane productions (**a**) pretreated switchgrass at 3% SL, 100 °C, 2% Ca(OH)_2_, 16 h; (**b**) pretreated switchgrass at 3% SL, 50 °C, 0% Ca(OH)_2_, 6 h; (**c**) pretreated switchgrass at 3% SL, 100 °C, 0% Ca(OH)_2_, 6 h and (**d**) raw switchgrass.

**Table 1 molecules-27-06891-t001:** Characterization analysis results of Shawnee.

Component	Result	Std. Dev.
**Solid**		
Total Solid, TS (g kg^−1^ Sample)	938.12	0.54
Volatile Solid, VS (g kg^−1^ Sample)	824.31	3.36
**Van Soest Fractions**		
Neutral Detergent Solubles (%)	26.54	0.48
Hemicellulose (%)	34.76	0.36
Cellulose (%)	33.13	0.44
Acid Detergent Lignin (%)	5.57	0.39
**Elemental Analysis**		
Carbon, C (%)	40.13	
Hydrogen, H (%)	5.75	
Nitrogen, N (%)	0.87	
Sulfur, S (%)	-	
**Total Structural Carbohydrates (%)**	52.86	0.559
Cellobiose (%)	0	
Glucose (%)	29.41	0.347
Xylose (%)	17.36	0.167
Galactose (%)	3.23	0.018
Arabinose (%)	2.87	0.027
Mannose (%)	0	

**Table 2 molecules-27-06891-t002:** Model equation and statistics for biochemical methane potential (BMP).

Biochemical Methane Potential Model (Modified, Reduced Quadratic)					
Source	Sum of Squares	Degree of Freedom	Mean Square	F-Value	*p*-Value
** *Model* **	83,292.64	6	13,882.11	19.01	<0.0001
*B-Solid loading (SL, wt.%) as DM*	43,543.01	1	43,543.01	59.63	<0.0001
*C-Reaction time (h)*	65.62	1	65.62	0.0899	0.7676
*D-Ca(OH)_2_ conc. (wt.%)*	3625.80	1	3625.80	4.97	0.0381
*BD*	4480.47	1	4480.47	6.14	0.0228
*B* ^2^	5584.90	1	5584.90	7.65	0.0123
*C* ^2^	5819.11	1	5819.11	7.97	0.0109
*Residual*	13,874.91	19	730.26		
*Lack of Fit*	13,260.18	18	736.68	1.20	0.6269
*Pure Error*	614.73	1	614.73		
*Cor Total*	97,167.54	25			
** *Fit Statistics* **					
*Standard Deviation (Std. Dev.)*	27.02		*R* ^2^		0.8572
*Mean*	134.21		*Adjusted R* ^2^		0.8121
*Coefficient of Variation (CV)*	20.13		*Predicted R* ^2^		0.7646
			*Adequate Precision*		10.7861
** *Model Equation in Terms of Coded Factors* **					
*(mL CH*_4_*gVS^−1^)* = *+76.47 − 49.18 × B − 1.91 × C + 14.19 × D + 16.73 × BD + 41.28 × B*² *+ 42.13 × C*²					

**Table 3 molecules-27-06891-t003:** Pretreatment conditions for switchgrass and results available in the literature.

Applied Pretreatment	Pretreatment Conditions	Results	Reference
Ca(OH)_2_ Assisted Thermal	Chemicals: Ca(OH)_2_: 0–2% Solid Loading: 3–7% Reaction Temperature: 50–100 °C Reaction Time: 6–16 h	BMP Raw Switchgrass (Shawnee): 217.1 mL CH_4_ gVS^−1^ CH_4_ Yield at Optimized Conditions: 248.7 mL CH_4_ gVS^−1^ at 3% solid loading as DM, 0% Ca(OH)_2_, 100 °C, 6 h. Enhancement Compared to Raw Switchgrass: 14.5%	This Work
H_2_O_2_ and Acetic Acid (Hac) Assisted Thermal	Chemicals: H_2_O_2_: 0–2% and Hac: 0–2% Reaction Temperature: 50–100 °C Reaction Time: 6–24 h	BMP Raw Switchgrass (Shawnee): 195.5 mL CH_4_ gVS^−1^ CH_4_ Yield at Optimized Conditions: 342.63 mL CH_4_ gVS^−1^ at 1.87% Hac, 0% H_2_O_2_, 50 °C, 6 h. Enhancement Compared to Raw Switchgrass: 75.2%	[3]
H_2_O_2_ Assisted Thermal	Chemicals: H_2_O_2_ 1–3% Solid loading: 3–7% Reaction Temperature: 50–100 °C Reaction Time: 6–24 h	BMP Raw Switchgrass: 208.4 mL CH_4_ gVS^−1^ CH_4_ Yield at Optimized Conditions: 291.34 mL CH_4_ gVS^−1^ at 6.43% solid loading, 1.83% H_2_O_2_, 50 °C, 6.78 h. Enhancement Compared to Raw Switchgrass: 39.8%	[4]
Chemical Pretreatment (NaOH, KOH, Ca(OH)_2_, H_2_O_2_, HCl, H_2_SO_4_) Steam Explosion	Chemical Pretreatment Chemicals: NaOH, KOH, Ca(OH)_2_, H_2_O_2_, HCl, H_2_SO_4_ 2, 3, 4, 5% *w*/*v*. Reaction Temperature: 25 °C Reaction Time: 12 h Solid loading: 8% (200 g Switchgrass/2.5 L) Steam Explosion Pressure: 1.2, 1.5, 1.8 Mpa Time: 20 Min.	BMP Raw Switchgrass: 46.3 mL CH_4_ gVS^−1^ CH_4_ Yield at Optimized Conditions: 197.2 mL CH_4_ gVS^−1^ at 4% NaOH, 25 °C, 12 h. Enhancement Compared to Raw Switchgrass: 325.9% CH_4_ Yield from Pretreated with 5% Ca(OH)_2_: 90.0 mL CH_4_ gVS^−1^ The effect of Ca(OH)_2_ pretreatment was not desirable compared to NaOH and KOH pretreatments.	[31]
Microwave Pretreatment	Final Reaction Temperature: 100, 150, 180 °C Reaction Time: 0–10–20 min. Temperature Increase: 5, 7.5, 10 °C/min	BMP Raw Switchgrass (Alamo): -mL CH_4_ gVS^−1^ CH_4_ Yield of Switchgrass Leaf at Optimized Conditions: 134.81 mL CH_4_ gVS^−1^ at 100 °C, 10 min, and 7.5 °C/min. Enhancement Compared to Raw Leaf Switchgrass: 9.1% CH_4_ Yield of Switchgrass Stem at Optimized Conditions: 99.35 mL CH_4_ gVS^−1^ at 150 °C, 10 min, and 10 °C/min. Enhancement Compared to Raw Stem Switchgrass: 5.2%	[32]
Chemical and Enzymatic Pretreatment	Chemical Pretreatment Chemicals: NaOH 1% (*w*/*v*) Liquid: Solid Ratio:10:1 Reaction Temperature: 50 °C Reaction Time: 12 h Enzymatic Pretreatment Reaction Time: 72 h Reaction Temperature: 50 °C Enzymes: Novozyme^®^188 (Cellobiase from *Aspergillus niger*) 35 FPU Celluclast^®^1.5 L (Cellulase from *Tricho-derma reesei* ATCC 26921) 61.5 CBU	BMP Raw Switchgrass (Kanlow): 197.39 mL CH_4_ gVS^−1^ CH_4_ Yield of Chemically Pretreated Switchgrass: 255.35 mL CH_4_ gVS^−1^ Enhancement Compared to Raw Switchgrass: 29.4% CH_4_ Yield of Chemically and Enzymatically Pretreated Switchgrass: 373.03 mL CH_4_ gVS^−1^ Enhancement Compared to Raw Switchgrass: 89%	[24]
Low Heat and Chemical Pretreatment	Chemicals: NaOH, Ca(OH)_2_, H_2_O_2_ 0, 2.2, 5.5, 11, 22% NaOH, 6.6%H_2_O_2_ Reaction Temperature: Room Temp—100 °C Reaction Time: 3, 6, 24 h	BMP of Fine Grind Raw Switchgrass (Cave-in-Rock): 296 mL CH_4_ gVS^−1^ CH_4_ Yield of Switchgrass at Optimized Condition: 332 mL CH_4_ gVS^−1^ at 5.5% NaOH, 100 °C, 6 h. Enhancement Compared to Raw Switchgrass: 12.2%	[27]
Microwave Pretreatment	Reaction Temperature: 90–180 °C Reaction Time: 7.5–32.6 Min.	BMP of Fine Grind Raw Switchgrass (Kanlow): 296 mL CH_4_ gVS^−1^ CH_4_ Yield of Switchgrass at Optimized Condition: 320 mL CH_4_ gVS^−1^ at 150 °C Enhancement Compared to Raw Switchgrass: 8.1% Microwave pretreatment induced no significant effect on methane production.	[25]

**Table 5 molecules-27-06891-t005:** Predicted methane production and kinetic parameters from the models.

**Model**	**Parameter**	**Switchgrass**			
		**Pretreated at**	**Pretreated at**	**Pretreated at**	**Raw**
		**3% SL, 100 °C, 2% Ca(OH)_2_, 16 h**	**3% SL, 50 °C, 0% Ca(OH)_2_, 6 h**	**3% SL 100 °C, 0% Ca(OH)_2_, 6 h**	
	** *M (mL CH_4_ gVS^−1^_added_)* **	176.9	164.05	178.26	199.31
**First Order**	** *k (d^−1^)* **	0.159	0.115	0.122	0.145
	** *P_predicted_ (mL CH_4_ gVS^−1^_added_)* **	173.92	161.68	170.88	188.71
	** *R* ^2^ **	0.988	0.994	0.992	0.985
	** *Adjusted R* ^2^ **	0.986	0.993	0.991	0.984
	** *Difference (%)* **	1.68	1.45	4.14	5.31
**Cone**	** *k (d^−1^)* **	0.21	0.136	0.165	0.183
	** *n* **	1.109	1.328	1.303	1.341
	** *P_predicted_ (mL CH_4_ gVS^−1^_added_)* **	169.5	159.87	166.82	188.9
	** *R* ^2^ **	0.993	0.997	0.992	0.983
	** *Adjusted R* ^2^ **	0.992	0.996	0.991	0.982
	** *Difference (%)* **	4.18	2.55	6.41	5.22
**Modified Gompertz**	** *R_m_ (mL CH_4_ gVS^−1^_added_ d^−1^)* **	17.44	14.25	14.37	15.97
	** *λ (day)* **	0.011	0.064	0.03	0.023
	** *P_predicted_ (mL CH_4_ gVS^−1^_added_)* **	171.34	154.17	169.89	192.51
	** *R* ^2^ **	0.954	0.984	0.967	0.976
	** *Adjusted R* ^2^ **	0.949	0.982	0.963	0.973
	** *Difference (%)* **	3.14	6.02	4.69	3.41
**Reaction Curve**	** *R_m_ (mL CH_4_ gVS^−1^_added_ d^−1^)* **	26.33	20.19	20.33	22.7
	** *λ (day)* **	0.018	0.059	0.04	0.038
	** *P_predicted_ (mL CH_4_ gVS^−1^_added_))* **	178.3	160.67	174.4	199.71
	** *R* ^2^ **	0.99	0.994	0.992	0.986
	** *Adjusted R* ^2^ **	0.989	0.994	0.992	0.984
	** *Difference (%)* **	−0.79	2.06	2.16	−0.2

**Table 6 molecules-27-06891-t006:** Effects of pretreatments on methane production kinetic parameters available in the literature.

Sample	Kinetic Parameters	Results			Reference
Switchgrass s-CSTR	First Order (FO) Cone (C) Modified Gompertz (MG) Reaction Curve (RC)	FO: k: −0.046–0.054 d^−1^ C: *n*: 1.7–1.9 MG: *R_m_*: 7.1–9.3 mL CH_4_ gVS^−1^ d^−1^ RC: *R_m:_* 9.45–12.6 mL CH_4_ gVS^−1^ d^−1^			[26]
The Olive Pomace Alkaline Pretreatment (NaOH: 2%, 4% and 8% (*w*/*w* TS)) Microwave NaOH + Microwave	Modified Gompertz	Untreated Raw: *R_m_*: 21.8 mL CH_4_ gVS^−1^ d^−1^ λ: 3.8 d NaOH Pretreatment: *R_m_*: 22.4–50.9 mL CH_4_ gVS^−1^ d^−1^ λ: 0–5.6 d			[7]
Switchgrass Chemical Pretreatment (NaOH, Ca(OH)_2_)	Modified Gompertz	Sample	μ_m_ (mL CH_4_ gVS^−1^ d^−1^)	λ (d)	[31]
		Untreated SG	1.9	3.1	
		4% NaOH	17.9	2	
		5% Ca(OH)_2_	6.3	0.8	
Switchgrass Microwave Pretreatment (90–180 °C)	First Order	k: 0.080–0.134 The increase in the pretreatment temperature caused an increase in the k coefficient, except for 105 °C. The SG biodegradability accelerated. The 105 °C pretreatments showed similar results to the raw sample.			[25]
Microwave Pretreatment Leaf and Stem fraction	First Order	Leaf–k: 0.021–0.075 d^−1^ Stem–k: 0.021–0.070 d^−1^ The k values were increased by 44% and 68% at 150 °C and 180 °C, respectively, compared with the control.			[32]

## Data Availability

Not applicable.

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
