# Peer review of "Energy Crops and Methane: Process Optimization of Ca(OH)2 Assisted Thermal Pretreatment and Modeling of Methane Production"

_molecules, 2022, doi:10.3390/molecules27206891_

Round 1

Reviewer 1 Report

This study attempted to demonstrate the Ca(OH)2 pretreatment applicable for enhancing biomass enzymatic saccharification in bioenergy crop. As this chemical is relatively low-cost and recyclable, it has been applied as a green-like and cost-effective technology. Although this study showed a very limited enhancement of biomass enzymatic saccharification, it obviously missed to cite more-recently-published articles, leading to the lack of essential comparison with other CaO pretreatments in previous studies. Below, my points should be well addressed in the revised manuscript, prior to being considerable for publication in this suitable journal:

Main Points:

1.         Since CaO pretreatments have been well performed with diverse lignocellulose substrates in major bioenergy crops, the revised manuscript should briefly describe previous studies in the “Introduction” section and add those major data into the Table 3 to see the advantage and disadvantage of this study by cite more-recently-published or more-related references such as:  (1) Madadi M., 2021, Using Amaranthus green proteins as universal biosurfactant and biosorbent for effective enzymatic degradation of diverse lignocellulose residues and efficient multiple trace metals remediation of farming lands, Journal of Hazardous Materials, 406: 124727; (2) Wu L. et al., 2019, Altered carbon assimilation and cellulose accessibility to maximize bioethanol yield under low-cost biomass processing in corn brittle stalk, Green Chemistry, 21: 4388; (3) Hu M. et al., 2018, Distinct polymer extraction and cellulose DP reduction for complete cellulose hydrolysis under mild chemical pretreatments in sugarcane, Carbohydrate Polymers. 202: 434-443; (4) Jin W. et al., 2016, Tween-80 is effective for enhancing steam-exploded biomass enzymatic saccharification and ethanol production by specifically lessening cellulase absorption with lignin in common reed, Applied Energy, 175: 82-90.

2.         As a classic alkali pretreatment with Ca(OH)2 , to our knowledge, it should mainly extract lignin with minor co-extraction of hemicellulose and non-crystalline cellulose, leading to a reduced cellulose DP (degree of polymerization) for much raised cellulose surface accessibility in the pretreated lignocellulose residues, which should be introduced and discussed in the revised manuscript by citing the references mentioned above.

Minor points

3.         “Title” looks like a review article, and it should highlight its major findings and novelty.
4.  At Table 1, what is the “Cellobiase”? It should be Cellubiose?

5.       At Figure 4,  it is hard to understand the optimal models by 3D RSP. 

6.       It cited too many very old and unessential references, and most of them should be deleted. In particular, some chemical analysis methods should cite more-recently-published references that have improved the methods applied.

Author Response

First, we would like to thank the reviewers for investing their valuable time in reviewing our article. We appreciate the careful reading and valuable comments of the reviewers and editor to help us improve this manuscript. Our responses to the reviewer’s questions can be found in attached file. 

Reviewer 2 Report

1.     Please move the Materials and Methods section before the Results and Discussion section.

2.     Line 435-437: How long was biomass kept in plastic boxes? temperature? Please add these information.

3.     Figure 6 is hardly legible. Please correct.

Author Response

(The authors gave the same response as above.)

Reviewer 3 Report

The article deals with the optimization of the Ca(OH)2 assisted thermal pretreatment of switchgrass to enhance methane production. The paper shows interesting studies with applicable lignocellulosic energy crop, an important topic in renewable energy field. Practical and kinetic modeling described in this work additionally increases its scientific value. However, it is not in a publishable shape. To improve the manuscript and to consider it worthy of publication, there are some critical issues to be checked and improved. 

I)                   First, it is necessary to clarify the manuscript better. The English expression of the is awful and needs to be revised. I needed to re-read the manuscript a few times to understand partially. If I understood correctly, the work deals mainly with Ca(OH)2 pretreatment of switchgrass for methane production, but, in some results, the authors discuss the general pretreatment of switchgrass. What is supposed to mean? It is not at all clear and can lead the reader to misunderstand. Pretreated switchgrass composition depends on technology and the severity of conditions used. Furthermore, the title of the manuscript should include "pretreatment of switchgrass for methane production."

II)                In the same line, in some parts, the introduction is too much of a collection of loose facts and too little background about the subject of the work. Thus, it is not at all clear and can lead the reader to misunderstand. I suggest that the authors should take care to present a proper background and argument to improve the manuscript, especially thinking about the journal's audience.

III)              The materials & methods section does not describe how the experiments were done in detail. Indeed, I would say that, in some cases, it does not describe the procedure. The results' interpretation is impaired if the materials and methods section does not make this sufficiently clear (it practically does not give complete information).  

IV)             Please check the size and resolution of the figures. It was hard to review and analyze them.

V)                Please check and provide an appropriate footnote to the figures and tables. It is not clear to the reader the different abbreviations used. The reader does not understand the elements (no description).

In addition to this general criticism, more detailed comments:

Please check the correct definition, classification, and form of the expression of the biochemical terms throughout the manuscript, such as Ssugar,  C3 and C4 photo-synthesis; biochemical hydrolysis? structural carbohydrate ...?

Please check the correct definition of the SI units throughout the manuscript.

In lines 26-32: Is there an appropriate consensus or references to support this claim?

In line 51: reference? Please see:

https://doi.org/10.1007/s13399-022-02981-5

https://doi.org/10.3390/biology10121277

https://doi.org/10.1016/j.isci.2022.104610

In line 83:  What supposed to mean "The general composition of switchgrass Shawnee is presented in  Table 1, and detailed results were discussed in BaÅŸar et al.”? Are you talking about BaÅŸar et al. results?

In line 134: In a footnote to the table and figures, you need to explain what the errors represent. Are they standard errors? If so, “of the sample”? Or “of the mean”?

In lines 120-132: Sorry, but  I needed to re-read the manuscript a few times to understand what was done partially. It is unclear how the experiments were done: Was used HPLC or DNS for sugar analysis? What is this supposed to mean? Unfortunately, the materials and methods section does not make this sufficiently clear (it essentially does not give any more information) for the reader to understand!

In line 277. the message that the authors want to pass is not clear. There is a mixture of information and poorly constructed sentences without a proper argument: What is the importance of the analysis? Please see the reference:

 Int. J. Mol. Sci. 202122(17), 9445; https://doi.org/10.3390/ijms22179445

In line 308: In the FTIR spectra,  I did not understand the comparison of the spectra.   It was hard to review and analyze them. Again, it is hard to understand the real significance of what the authors have done and the interpretation of the results.

Author Response

(The authors gave the same response as above.)

Round 2

Reviewer 3 Report

All revisions were made by the author.